# Mood Disorders in Levothyroxine-Treated Hypothyroid Women

**DOI:** 10.3390/ijerph16234776

**Published:** 2019-11-28

**Authors:** Benjamín Romero-Gómez, Paula Guerrero-Alonso, Juan Manuel Carmona-Torres, Blanca Notario-Pacheco, Ana Isabel Cobo-Cuenca

**Affiliations:** 1Hospital El Tomillar de Sevilla, Servicio Andaluz de Salud (SAS), 41500 Alcalá de Guadaira, Spain; volvart@hotmail.com; 2Centro de Salud Najera, Servicio Rioja Salud, 26300 Najera, Spain; pguerrero@riojasalud.es; 3Facultad de Fisioterapia y Enfermería y Fisioterapia de Toledo, Universidad de Castilla la Mancha, 45005 Toledo, Spain; anaisabel.cobo@uclm.es; 4Grupo de Investigación Multidisciplinar en Cuidados, Universidad de Castilla la Mancha, 45005 Toledo, Spain; 5Instituto Maimónides de Investigación Biomédica de Córdoba (IMIBIC), 14004 Córdoba, Spain; 6Facultad de Enfermería de Cuenca, Universidad de Castilla la Mancha, 16071 Cuenca Toledo, Spain; blanca.notario@uclm.es; 7Grupo de Investigación CESS, Universidad de Castilla la Mancha, 16071 Cuenca, Spain

**Keywords:** hypothyroidism, mood disorders, levothyroxine, women

## Abstract

*Background*: Hypothyroidism has several symptoms (weight gain, arrhythmias, mood changes, etc.). The aims of this study were (1) to assess the prevalence of anxiety and depression in levothyroxine-treated hypothyroid women and in women without hypothyroidism; (2) to identify variables associated with anxiety and depression. *Methods*: A case-control study was performed with 393 women. Case-group: 153 levothyroxine-treated hypothyroid women. Control-group: 240 women without hypothyroidism. Convenience sampling. Instrument: The Hamilton Hospital Anxiety and Depression Scale (HADS), and a sociodemographic questionnaire. *Results*: The prevalence of anxiety in levothyroxine-treated hypothyroid women was higher than in women without hypothyroidism (29.4% vs. 16.7%, χ^2^*p* < 0.001). The prevalence of depression in the case group was higher than in the control group (13.1% vs. 4.6%, χ^2^
*p* < 0.001). Levothyroxine-treated hypothyroid women were more likely to have anxiety (OR = 2.08, CI: 1.28–3.38) and depression (OR = 3.13, IC = 1.45–6.45). *Conclusion*: In spite of receiving treatment with levothyroxine, women with hypothyroidism are more likely to have depression and anxiety. Health professionals need to assess the mood of women with hypothyroidism. Although levothyroxine is a good treatment for the symptoms of hypothyroidism, it may not be enough to prevent development or persistence of depression and anxiety by itself.

## 1. Introduction

Hypothyroidism is the most common thyroid dysfunction in the population [1]. Its prevalence is usually around 1%–2% of the population, although this rate increases with age. It is up to 10 times more common in women than in men [2]. In Spain, different studies indicate prevalence ranging from 1.36% to 3.95% [1,3].

Hypothyroidism has been associated with different neuropsychiatric disorders (mania, acute psychosis, psychiatric disturbances due to iatrogenic hypothyroidism, cognitive disorders, and, primarily, mood changes) [4,5,6,7,8,9,10,11]. A higher prevalence of depressive symptoms and worse scores on depression scales have been found in patients with clinical and subclinical hypothyroidism [11,12,13,14], and a higher prevalence of hypothyroidism has been found in patients with major depressive disorders (MDD) [15,16,17,18]. Regarding anxiety, its prevalence in hypothyroid patients can reach 63%–65% [7,11,13,19]. Different authors also found a relationship between scores on anxiety scales and hypothyroidism [4,20,21].

The usual treatment for clinical hypothyroidism is levothyroxine sodium monotherapy [22]. Its use has shown an evident reduction in symptomatology, including depression [23,24]. It also makes it possible to normalize the levels of thyroid hormones and thyroid-stimulating hormone (TSH) serum [25,26]. The dose may vary due to various circumstances such as the appearance of symptoms or side effects, comorbidity, significant alterations in weight, the use of estrogens, being pregnant, age, etc. [22,27]. Not all hypothyroid patients seem to improve in the same way with this treatment. Occasionally, patients continue to present symptoms despite having reached euthyroidism [4,28,29].

The effect of levothyroxine treatment on anxiety and depression in patients with clinical hypothyroidism is an issue that continues to raise questions. On the one hand, Wiersinga (2017) reported that between 5% and 10% of hypothyroid patients treated with levothyroxine had mood symptoms despite having reached euthyroidism [6]. Working with 130 levothyroxine-treated hypothyroid patients, Djurovic et al. (2018) found significant differences in the Hamilton Hospital Anxiety and Depression Scale (HADS) scores in depression and anxiety (in patients under 50 years of age) [30]. Those effects were accentuated if they had Hashimoto’s thyroiditis [30]. Yalcin et al. (2017) showed that Hashimoto patients had significant differences in the Beck Depression Inventory-II (BDI-II) scores compared to those of the control group, although their patients had been receiving levothyroxine treatment for only 3 months [31]. Finally, Panicker et al. (2009) found that women who took levothyroxine and were euthyroid had a higher prevalence of anxiety and depression than women in the control group [20].

On the other hand, Gulseren et al. (2006) found that, after levothyroxine treatment, anxiety and depression levels in a group of 33 patients with clinical hypothyroidism were lower than in the control group [32]. In a sample of 1503 patients, Medici et al. (2014) found that low TSH levels (0.3–1.0 mIU/L) were associated with more depressive symptoms and more depression than high TSH levels (1.6–4.0 mIU/L) [33]. Finally, Ittermann (2015) only found a link between thyroid disorders and BDI-II C12 scores if treated patients were excluded [4].

The efficacy of levothyroxine on reducing symptoms of anxiety and depression in patients with clinical hypothyroidism remains unclear, and there are differences between the studies carried out. Moreover, in the case of our country, there are no similar studies on the subject. Given these data, the aims of our study were (1) to evaluate the levels of anxiety and depression in levothyroxine-treated hypothyroid women, (2) to compare the mood (anxiety and depression) of levothyroxine-treated hypothyroid women and women without hypothyroidism, and (3) to assess the influence of different variables on mood.

## 2. Materials and Methods

### 2.1. Participants and Design

From September 2018 to March 2019, a multicenter case-control study was carried out in two Spanish regions. Five hundred and five women with and without hypothyroidism and over 18 years of age were recruited among the patients of two Primary Care Health Zones located in central and southern Spain.

The inclusion criteria for the case group (women with primary hypothyroidism) were a diagnosis of primary hypothyroidism treated with levothyroxine for at least 6 months, and normalized TSH levels (0.4–4.0 mIU/L). The exclusion criteria were subclinical hypothyroidism diagnosis, high serum TSH despite replacement therapy, being pregnant or lactating, having serious psychiatric pathologies (e.g., schizophrenia) and the consumption of medications or substances (e.g., antipsychotics, parenteral drugs) that can affect mood.

The inclusion criterion for the control group (women without hypothyroidism) was an absence of thyroid disorders (clinical or subclinical hypothyroidism, clinical or subclinical hyperthyroidism). The exclusion criteria were having a family history of clinical or subclinical hypothyroidism, having abnormal TSH levels, and also the exclusion criteria of the case group.

### 2.2. Sample Size

Granmo software (version 7.12, Barcelona, Spain, 2012) was used for the sample size. In a similar study, Djurovic et al. (2018) [30] reported a mean anxiety score in levothyroxine-treated hypothyroid women of 11.07 ± 3.17 and of 9.65 ± 3.07 in control group patients. We have not found similar studies in the Spanish population. Therefore, for a 2-point difference in the HADS anxiety subscale scores, with an accuracy of 5%, a 95% confidence interval, and an estimated loss rate of 20%, 50 women were needed in each group.

### 2.3. Instruments

An online questionnaire was constructed that included different instruments to measure the variables:(1)Zigmond and Snaith’s Hospital Anxiety and Depression Scale (HADS) [34] in the Spanish version by Caro e Ibáñez [35] is a self-administered questionnaire that assesses the extent of anxiety and depression in non-psychiatric patients. It consists of 14 questions with 4 response options grouped into 2 subscales, anxiety (odd items) and depression (even items). The answers for each item are scored from 0 to 3 points for a total of 21 on each subscale. The cut-off points are the same for anxiety and depression and are Normal (0–7 points), Doubtful (8–10 points), or Clinical Problem (11 or more points). It has high internal consistency (Cronbach alpha 0.84–0.87) and an adequate convergent validity (*p* < 0.005). Psychometric properties confirm that they provide a reliable measure of anxiety and depression among the Spanish population [36,37,38].(2)A questionnaire on sociodemographic variables: age, educational level, employment status, alcohol consumption, tobacco consumption, substance consumption, civil status, and cohabitation with a partner.(3)Clinical variables: Body Mass Index (BMI) (weight [kg]/height [m^2^]). Weight and height were measured at the time of the interview. For levothyroxine-treated hypothyroid women, the TSH and the daily intake of levothyroxine were collected.

### 2.4. Procedure

Convenience sampling was used. Primary care professionals from both health areas were contacted to collaborate in the study. During annual reviews, the physicians selected and informed those women who met the inclusion and exclusion criteria for each group (case and control) in this study. Women who chose to participate were given an informed consent form to complete and the link to the questionnaire. The questionnaire did not collect personal data or any other type of data unrelated to the variables of the study.

### 2.5. Study Variables

#### 2.5.1. Independent Variables

The independent variables were presence/absence of hypothyroidism (dichotomous), TSH levels (quantitative), levothyroxine intake (categorical), age (quantitative/categorical), BMI (quantitative/categorical), educational level (categorical), employment status (categorical), civil status (categorical), cohabitation (categorical), alcohol consumption (dichotomous), tobacco consumption (dichotomous), substance consumption (categorical).

#### 2.5.2. Dependent Variables

The dependent variables were the anxiety and depression scores collected through the HADS questionnaire. These variables were used as quantitative and categorical variables.

### 2.6. Ethical Considerations

This work follows the fundamental principles of the UNESCO Universal Declaration of Human Rights, the Helsinki Declaration, and Spanish Organic Law 15/1999, of 13 December, on the Protection of Personal Data, and Royal Decree 994/99, of 11 June, of the Spanish State. The study was approved by the institutional ethical committees (the Ethical Investigation Committee of Toledo (CEIC TO-2019) and the Ethical Investigation Committee of Málaga (CEICMa-2019).

## 3. Results

### 3.1. Clinical and Sociodemographic Variables

Of the 505 women who participated, 393 completed the survey correctly. These women were divided into 2 groups: 153 levothyroxine-treated hypothyroid women and normalized TSH (case group, 38.93%) and 240 non-hypothyroid women (control group, 61.07%). The mean age of the participants was 35.38 ± 9.46 years with a BMI of 23.55 ± 4.12. There were 285 (72.5%) employed women and 275 (70%) women were married. The majority (59%) lived with their partners. The mean TSH level of levothyroxine-treated hypothyroid women was 2.43 ± 1.07.

Clinical and sociodemographic variables of both groups (Table 1) were compared using the Student’s t test (quantitative) and the χ-square test (categorical).

### 3.2. Anxiety and Depression

Significant differences in anxiety and depression were found when comparing both groups (Table 2). The women in the case group scored significantly worse on the anxiety (*p* < 0.001) and depression (*p* < 0.001) subscales. The prevalence of anxiety (*p* = 0.003) and depression (*p* < 0.001) were also significantly higher in levothyroxine-treated hypothyroid women.

Using logistic regression, the influence of the independent variables on anxiety (Table 3) and depression (Table 4) was analyzed. The BMI variables (normal weight, overweight, and obesity) and age (<30, 30–39, 40–49, and 50+) were categorized to facilitate the analysis. HADS scores were also categorized into two groups: 0–10 (Normal) and 11+ (with a clinical problem), in line with different authors [38].

Levothyroxine-treated hypothyroid women with normalized TSH (case group) showed an increased risk of anxiety disorders (*p* = 0.003; OR: 2.08 [1.28–3.38]) compared to the control group (Table 3).

In relation to age, the 40–49 age group had an increased risk of anxiety in the control group (*p* = 0.035; OR: 2.61 [1.07–6.38]). There were no differences in the other groups (Table 3).

BMI increased the risk of anxiety in the case group (*p* = 0.007; OR: 3.03 [1.36–6.78]) and in the total sample (*p* = 0.003; OR: 2.37 [1.34–4.20]) if the women were overweight. This effect was not found in women with hypothyroidism (Table 3).

No other variables showed any influence on anxiety.

The logistic regression found that hypothyroidism was a risk factor for depressive disorders (*p* = 0.004; OR: 3.13 [1.45–6.73]) (Table 4).

In terms of age, women over 50 in the total sample had a higher risk of depression (*p* = 0.026; OR: 3.47 [1.16–10.35]). This data was not found separately in the case and control groups.

BMI showed a clear influence on depression scores only in the control group (non-hypothyroid women); the risk increased if they were overweight (*p* = 0.023; OR: 5.25 [1.25–22.01]) or obese (*p* = 0.002; OR: 13.80 [2.71–70.19]). In the case of the total sample, this effect was only visible when the women were overweight (*p* = 0.012; OR: 2.85 [1.26–6.45]) (Table 4).

Of the remaining variables, only civil status influenced the risk of depression, with the risk being greater in the total sample when the participants were divorced (*p* = 0.008; OR: 6.20 [1.60–24.01]) (Table 4).

The specific influence of hypothyroidism on anxiety and depression was evaluated using analysis of covariance (ANCOVA), controlling the Age and BMI variables. Hypothyroidism continued to show a significant effect on anxiety and depression (*p* < 0.001) both when adjusted for age (Model 0) and when adjusted for age and BMI (Model 1) (Table 5).

Finally, Pearson’s correlation analysis found no association between TSH and depression (*p* = 0.637), or TSH and anxiety (*p* = 0.390) (Table A1).

## 4. Discussion

Treatment with levothyroxine reduces the classic symptoms of hypothyroidism (constipation, dry skin, fatigue, bradycardia, etc.). In this study, we have evaluated the influence of levothyroxine treatment on symptoms associated with anxiety and depression in women with hypothyroidism. Contrary to what was found by Gulseren (2006) [32], the mean anxiety and depression scores in our study were significantly higher in women with hypothyroidism and euthyroidism than in women in the control group. These significant differences persist even when we adjusted for the effects of age and BMI. Kelderman (2015) and Djurovic (2018), who also used the HADS scale, came up with similar results [31,39]. Yalcin (2017) [31] and Krysiak (2016) [40], who used the BDI and the Beck Anxiety Inventory (BAI), also found a similar association between treated hypothyroidism and anxiety and depression.

In our sample, 13.1% of levothyroxine-treated hypothyroid women had depression and 29.1% had anxiety. These figures were significantly higher than those of the control group; in fact, they almost tripled the prevalence of depression among women in the control group (13.1% vs. 4.6%). Panicker (2009) found a prevalence of depression in treated euthyroid women of 18.4% and of anxiety of 23.4% [20]; Ittermann (2015) found a prevalence of anxiety of 23.5% and of depression of 22.3% in women taking levothyroxine [4]. These figures are lower than our cut-off in anxiety (29.4%) but higher than the depression figures in our sample (13.1%). Finally, Giynas found prevalences of 33.3% (depression) and 37.3% (anxiety) in euthyroid patients with Hashimoto’s thyroiditis [41]. These three authors also agree with the data obtained in our study, that the prevalence of anxiety was greater than that of depression [4,20,41].

In our study, we found that women with hypothyroidism were 3.13 times more likely to suffer from depression than women without hypothyroidism and 2.37 times more likely to suffer from anxiety. Lin (2016) [23] and Giynas (2014) [41] also reported an increased risk of depression in patients with hypothyroidism, although the risk was reduced by taking levothyroxine [23]. Lin also points out that the risk of depression was higher in women [23]. Regarding anxiety, Benseñor (2015) found an increased risk of panic disorders in subjects with subclinical hypothyroidism [21].

Several studies have tried to explain the relationship between hypothyroidism and mood disorders. Many of the symptoms of clinical hypothyroidism overlap with the symptomatology of depression [6,42,43]. These include loss of appetite, loss of energy, sleep disturbances, tiredness, slower information processing, emotional lability, difficulty concentrating, memory and learning disturbances, etc. [7,44]. On the other hand, many patients who know they have hypothyroidism tend to report more symptoms (labelling effect). Panicker (2009) found an increased risk of depression in the subgroup of patients who knew they had been diagnosed with hypothyroidism [20], a circumstance similar to that found in other pathologies [45,46]. Difficulties in replicating the circadian physiological rhythm of thyroid hormone secretion have also been mentioned as another possible cause of these differences in mood [28,39].

At the brain level, low T3 levels have been found despite the presence of normal TSH levels. This has been related to polymorphism of deiodinase-2 [47,48] and mutations in the genes that regulate the transport of brain thyroid hormones [6,48,49]. The type 2 deiodinase converts T4 into T3 in the brain and the hypophysis, for instance [22]. Its polymorphism could affect the brain enzyme activity. It has been observed that individuals with the Thr92Ala deiodinase type 2 variant score worse in health questionnaires, even under levothyroxine treatment [49]. According to “hypothyroid brain” theory, mood alterations in these patients would be the consequence of a state of local hypothyroidism in the brain, with normal peripheral thyroid hormone levels [50,51]. Some of the markers that have been related to this “hypothyroid brain” are an increased thyrotropin releasing hormone (TRH) and low transthyretin (TTR) levels in the cerebrospinal fluid [51].

Different abnormalities have also been found in the hypothalamus–hypophysis–thyroid (HHT) axis in depressive patients [44]; in particular, those related to feedback between the HHT system and serotonin. Thyroid hormones regulate the sensitivity of serotonin receptors 5-HT1A y 5-HT1B, modifying serotonergic transmission [52]. Furthermore, an attenuated TSH response to TRH has been found in patients with depression [50]. There are those who have tried to apply similar explanations to the case of anxiety [20,53]. However, the final causes have not yet been established.

Several options have been studied in order to solve the persistency of the symptoms, for instance, combined T3 + T4 therapy. In a recent revision, Dayan (2018) reported that, at a population level, there is no benefit of combined T4 + T3 therapy over levothyroxine monotherapy [54], but its use is being considered for patients with persistent symptomatology. The use of this combined therapy is still under research.

In the women in our sample, no significant correlation was found between TSH and HADS scores. The relationship between TSH levels and depressive symptoms has not been clearly established. While certain studies [20,30,31,55,56] found a correlation between TSH elevation and depressive symptoms in hypothyroid women, Medici (2014) reported that, in fact, low TSH and elevated T4 were related to depressive symptoms [33]. On the other hand, different studies on general population found no relationship between TSH and anxiety symptoms [4,57].

In terms of age, women in the total sample over 50 were 3.47 times more likely to have depression. Non-hypothyroid women aged 40–49 were 2.61 times more likely to have anxiety. Several studies have associated menopause and postmenopause with an increase in depressive and anxiety symptoms [58,59,60]. The different hormonal changes associated with that period may be reflected in their mood, although it is striking that this effect is less noticeable in women with hypothyroidism. It is also interesting to note that the risk of anxiety increases during the beginning of the period of change and transition, whereas depression become more visible after the age of 50, when many women are already entering the postmenopausal stage. ANCOVA was used to isolate the effect of age and it was observed that it ceased to be significant between groups, which could be due to the fact that menopause is a phenomenon common to all women.

Regarding BMI, non-hypothyroid and overweight women were found to be 5.25 times more likely to have depression, and that risk increased 13.8 times if they were obese. These results are consistent with a recent meta-analysis, which showed that there was evidence of overweight or obesity being associated with depression in adult people [61]. In terms of anxiety, overweight increased the risk of anxiety in non-hypothyroid women (OR: 3.03) and in the total sample (OR: 2.37). Isolated from other variables, ANCOVA showed that the effect of BMI was constant and significant. Overweight and obesity have been associated with depressive symptoms [62] and have been observed to increase with age [63,64]. Although hypothyroid women had a higher BMI than women in the control group, BMI has not been found to increase the risk of depression or anxiety in this group. For future studies, it may be interesting to assess if this is due to a minor protective effect on hypothyroid women.

Civil status influenced the risk of depression. In our sample, divorced women were 6.20 times more likely to have depression, in line with Rosenström (2017) [65].

### Strengths and Limitations

The strengths of this study include, first of all, a larger sample than several of the studies with which it has been compared. Furthermore, to the extent of our knowledge, this is the first study on this topic in the Spanish population.

This study has certain limitations. First, as it is a case-control study, it is not possible to establish causal relationships. In addition, although the TSH level from their last analysis was reviewed in order to select the participants, this was only collected in the group of levothyroxine-treated hypothyroid women and normalized TSH (case group), so we have not been able to compare the TSH intergroup.

## 5. Conclusions

Despite being treated with levothyroxine and having reached euthyroidism, the prevalences of anxiety and depression in women with diagnosed hypothyroidism were higher than in women without hypothyroidism. These prevalences are not related to TSH levels when these fall within the reference range. An increase in BMI is significantly associated with an increase in depressive and anxious symptomatology, although this effect was not noticeable in levothyroxine-treated hypothyroid women.

## 6. Contributions of This Study

To the extent of our knowledge, this is the first research carried out on anxiety and depression in levothyroxine-treated hypothyroid women and normalized TSH in Spain.

This study demonstrates that, despite standard hypothyroidism treatment and normal TSH levels, mood disorders may persist. Although more studies of this type are needed, healthcare professionals should remain attentive to the impact of thyroid disorders on mood.

## Figures and Tables

**Table 1 ijerph-16-04776-t001:** Clinical and sociodemographic variables.

Variables	Case Group	Control Group	Total	Sig. (t)
	*n* = 153 (100%)	*n* = 240 (100%)	*N* = 393 (100%)	
Age	36.52 ± 9.96	34.66 ± 9.07	35.38 ± 9.46	0.058
Body mass index (BMI)	24.47 ± 4.62	22.96 ± 3.65	23.55 ± 4.12	<0.001
TSH	2.43 ± 1.07			**
Qualitative Variable				Sig. (χ^2^)
Educational level				<0.001
University	99 (64.7%)	194 (80.83%)	293 (74.6%)	
Secondary	39 (25.5%)	40 (16.17%)	79 (20.1%)	
Primary	15 (9.8%)	6 (2.5%)	21 (5.3%)	
Employment status				0.001
Employed	101 (66%)	184 (76.7%)	285 (72.5%)	
Student	17 (11.1%)	33 (13.8%)	50 (12.7%)	
Other	35 (22.9%)	23 (9.6%)	58 (14.8%)	
Civil status				0.028
Married/Steady Partner	118 (77.2%)	157 (65.4%)	275 (70%)	
Single	27 (17.6%)	71 (20.6%)	98 (24.9%)	
Divorced	8 (5.2%)	12 (5%)	20 (5.1%)	
Cohabitation				0.025
Yes	101 (66%)	131 (54.58%)	232 (59%)	
No	52 (34%)	109 (45.41%)	161 (41%)	
Tobacco Consumption				0.558
Yes	41 (26.8%)	58 (24.16%)	99 (25.2%)	
No	112 (73.2%)	182 (75.83%)	294 (74.8%)	
Alcohol Consumption				0.12
Yes	110 (71.9%)	189 (78.75%)	299 (76.1%)	
No	43 (28.1%)	51 (21.25%)	94 (23.9%)	
Substance Consumption				0.882
Yes	9 (5.9%)	15 (6.25%)	24 (6.1%)	
No	144 (94.1%)	225 (93.75%)	369 (93.9%)	
Levothyroxine intake				**
Over 100 µg	35 (27.5%)			
51–100 µg	76 (49.7%)			
50 µg	42 (22.8%)			
Etiology of hypothyroidism				
After surgery	18 (11.77%)			
Autoimmune thyroiditis	135 (88.23%)			

TSH: thyroid-stimulating hormone. ** Not comparable.

**Table 2 ijerph-16-04776-t002:** Hamilton Hospital Anxiety and Depression Scale (HADS) scores in case and control groups.

HADS Scores	Case Group	Control Group	Total	Sig (t)
	Mean Score	Mean Score		
Anxiety	8.86 ± 4.40	7.10 ± 3.72	7.79 ± 4.09	*p* < 0.001
Depression	5.79 ± 3.81	3.61 ± 3.34	4.46 ± 3.68	*p* < 0.001
	Prevalence	Prevalence		Sig (χ^2^)
Anxiety				
1–7	68 (44.4%)	144 (60%)	212 (53.9%)	
8–10	40 (26.1%)	56 (23.3%)	96 (24.4%)	0.003
11+	45 (29.4%)	40 (16.7%)	85 (21.6%)	
Depression				
1–7	105 (68.6%)	205 (85.4%)	310 (78.9%)	
8–10	28 (18.3%)	24 (10%)	52 (13.2%)	*p* < 0.001
11+	20 (13.1%)	11 (4.6%)	31 (7.9%)	

HADS: Hamilton Hospital Anxiety and Depression Scale (HADS).

**Table 3 ijerph-16-04776-t003:** Logistic regression model predicting anxiety and independent variables *.

Variables	Case Group		Control Group		Total Women	
	*p*	OR (CI 95%)	*p*	OR (CI 95%)	*p*	OR (CI 95%)
Age group						
>50	0.787	1.16 [0.38–3.57]	0.906	0.90 [0.18–4.57]	0.548	1.30 [0.54–3.12]
40–49	0.120	0.45 [0.17–1.22]	0.035	2.61 [1.07–6.38]	0.546	1.21 [0.64–2.31]
30–39	0.941	0.96 [0.40–2.32]	0.782	1.13 [0.46–2.74]	0.982	1.00 [0.55–1.84]
<30	Ref		Ref		Ref	
BMI						
Obesity (30+)	0.711	1.20 [0.44–3.27]	0.082	3.03 [0.86–10.63]	0.061	2.09 [0.96–4.53]
Overweight (25–29.9)	0.288	1.56 [0.68–3.54]	0.007	3.03 [1.36–6.78]	0.003	2.37 [1.34–4.20]
Normal weight (<25)	Ref		Ref		Ref	
Hypothyroidism						
Yes	-	-	-	-	0.003	2.08 [1.28–3.38]
No	-	-	-	-	Ref	-

* Only significant results are shown.

**Table 4 ijerph-16-04776-t004:** Logistic regression model predicting depression and independent variables *.

Variables	Case Group		Control Group		Total Women	
	*p*	OR (CI 95%)	*p*	OR (CI 95%)	*p*	OR (CI 95%)
Age group						
>50	0.056	3.69 [0.96–14.12]	0.707	1.56 [0.15–16.00]	0.026	3.47 [1.16–10.35]
40–49	0.504	0.60 [0.13–2.68]	0.377	2.00 [0.42–9.32]	0.835	1.11 [0.39–3.19]
30–39	0.778	1.20 [0.33–4.25]	0.848	0.84 [0.16–4.34]	0.983	1.01 [0.37–2.70]
<30	Ref		Ref		Ref	
BMI						
Obesity (30+)	0.274	0.31 [0.03–2.52]	0.002	13.80 [2.71–70.19]	0.215	2.07 [0.65–6.59]
Overweight (25–29.9)	0.338	1.64 [0.59–4.59]	0.023	5.25 [1.25–22.01]	0.012	2.85 [1.26–6.45]
Normal weight (<25)	Ref		Ref		Ref	
Hypothyroidism						
Yes	-	-	-	-	0.004	3.13 [1.45–6.73]
No	-	-	-	-	Ref	
Civil Status						
Divorced	0.052	7.50 [0.98–57.13]	0.121	4.53 [0.67–30.56]	0.008	6.20 [1.60–24.01]
Married	0.445	1.82 [0.39–8.48]	0.885	0.90 [0.21–3.70]	0.401	
Single						

* Only significant results are shown.

**Table 5 ijerph-16-04776-t005:** Mean differences (analysis of covariance [ANCOVA]) in anxiety and depression in case and control groups adjusted for age (Model 0) and adjusted for age + Body Mass Index (BMI) (Model 1).

Variables	Case Group		Control Group		*F*	*p*
Model 0	M(SD)	CI Adjusted	M(SD)	CI Adjusted		
Anxiety	8.86(0.32)	(8.22–9.50)	7.11(0.25)	(6.60–7.62)	17.65	<0.001
Depression	5.77(0.28)	(5.21–6.33)	3.63(0.22)	(3.18–4.08)	33.99	<0.001
Model 1	M(SD)	CI Adjusted	M(SD)	CI adjusted		
Anxiety	8.8(0.32)	(8.16–9.45)	7.14 (0.26)	(6.63–7.66)	15.44	<0.001
Depression	5.62(0.28)	(5.07–6.18)	3.72(0.22)	(3.28–4.16)	27.09	<0.001

M: marginal estimated means ± SD. Model 0: Adjusted for age. Model 1: Adjusted for age + BMI; Statistical significance (*p* ˂ 0.05) in pairwise mean comparisons using Bonferroni post-hoc test.

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
