# Peer review of "Mood Disorders in Levothyroxine-Treated Hypothyroid Women"

_ijerph, 2019, doi:10.3390/ijerph16234776_

Round 1

Reviewer 1 Report

The authors mentioned that participants had TSH levels (0.4-4-0 mIU/ml) and were consider without subclinical hypothyroidism. However, there are many studies indicating that a TSH level > 2.5 mIU/ml could be considered as subclinical hypothyroidism since many patients have symptoms and even antibodies that affect the thyroid function. Moreover, subclinical hypothyroidism has also been associated with anxiety and depression. In this way, a new selection of subjects considering a TSH < 2.5 could be done to the control group, normalized TSH group, and TSH < 2.5 group. This new analysis might give more information about the cut-off point to TSH to present anxiety and depression.

Also, it could be necessary to add information in the discussion section about the actions of thyroid hormones in the neurotransmitters and development of symptoms of anxiety and depression.

Author Response

Reviewer 1:

The authors mentioned that participants had TSH levels (0.4-4.0 mIU/L)  and were consider without subclinical hypothyroidism. However , there are many studies indicating that TSH level >2.5 mIU/L could be considered as subclinical hypothyroidism since many patients have symptons and even antibodies that affect the thyroid function. Moreover, subclinical hypothyroidism has also been associated with anxiety and depression. In this way, a new selection of subjects considering a TSH<2.5 could be done to the control group, normalized TSH group, and TSH<2.5 group. The new analysis might give more information about the cut-off point to TSH to prevent anxiety and depression.

First of all, thank you for your comments and suggestions to improve the manuscript. When the preselection of candidates was carried out, the main criterion was the presence of normal TSH levels defined according to the parameters of the American Thyroid Association Task Force on Thyroid Hormone Replacement. They establish that the TSH reference range must remain between 0.4–4.0 mIU/L [1]. In order to avoid bias, we considered that TSH should remain within this range also for women in the control group. It is true that there is controversy over the adequate reference range for clinical and subclinical hypothyroidism [2] and that some studies report a relation between subclinical hypothyroidism and mood disorders [3]. Nevertheless, the American Thyroid Association Task Force on Thyroid Hormone Replacement indicates that “[…] symptoms alone lack sensitivity and specificity and therefore are not recommended for judging adequacy of replacement in the absence of biochemical assessment. Therefore, symptoms should be followed but considered in the context of serum thyrotropin values, relevant comorbidities, and other potential causes.”[1] Since TSH levels were normal in women of the control group, we decided to base ourselves on the TSH data collected at the time of the selection.

In any case, we would like to be able to include that change. We agree that if offers even more information. However, it is impossible because, as we pointed out in the limitations section, TSH levels were not collected from controls; we examined the analysis of the last six months and verified the absence of new symptoms. The filters to preserve the anonymity of participants prevent us from matching data and subjects in order to conduct this comparison.

On the other hand, we believe that a possible case of subclinical hypothyroidism is not prejudicial to our study; in fact, these possible non-diagnosed cases would increase the prevalence of anxiety and depression in the control group [3]. Since our study shows significant differences between  levothyroxine-treated hypothyroid women and control women, removing a case of subclinical hypothyroidism would increase these differences. Nevertheless, it will be taken into account for future studies.

Also, it could be necessary to add information in the discussion section about the actions of thyroid hormones in the neurotransmitters and development of symptons of anxiety and depression

Following your suggestion, the information has been included.

References:

[1] Jonklaas, J.; Bianco, A.C.; Bauer AJ, Burman KD, Cappola AR, Celi FS, et al. Guidelines for the treatment of hypothyroidism: prepared by the american thyroid association task force on thyroid hormone replacement. Thyroid. 2014, 24(12), 1670–751.

[2] Chaker, L.; Bianco, A.C.; Jonklaas, J.; Peeters, R.P. Hypothyroidism. Lancet, 2017, 390, 1550–1562.

[3] Zhao, T.; Chen, B.B.; Zhao, X.M.; Shan, Z.Y. Subclinical hypothyroidism and depression: a meta-analysis. Transl Psychiatry, 2018, 8(1), 239.

Reviewer 2 Report

The authors presented an interesting issue of mood disorders, which occur in hypothyroid women despite a proper treatment with levothyroxine. However, I have several doubts which make it impossible to assess whether the obtained results are true for the analyzed groups. The inclusion criteria for the Control Group included “an absence of thyroid disorders (clinical or subclinical hypothyroidism, clinical or subclinical hyperthyroidism)” and the authors claimed that all patients with abnormal TSH level within the previous 6 months had been excluded. However, no data on the TSH or thyroid hormone levels are available for this group. Were any thyroid tests performed in this group? Did all of the patients really have normal TSH within 6 months before inclusion? Why weren`t the thyroid tests performed at the time of inclusion? How were the subclinical disorders assessed without the thyroid hormone levels? All these questions mean that the actual characteristics of the studied groups may be different from those assumed by the authors. It is impossible to obtain reliable results of the comparison without application of reliable diagnostic procedures allowing to clearly distinguish a group of healthy people and a case group. Moreover, the authors did not provide any data on the thyroid hormone levels in the analyzed case patients. There is only an information that the TSH levels were within the normal range and that only patients with treated overt hypothyroidism were included into the study. Mood disorders are frequently present in patients and many additional factors may influence the occurrence of depression and anxiety. The authors analyzed several of these factors but the results of the whole analysis cannot be accepted without the clear group selection.

If the authors are able to provide the lacking data and the group re-analysis, I will be ready to review the re-submission (after extensive language editing by a native speaker).

The Introduction should contain essential literature-based background for the study. The obvious basic knowledge should not be included [e.g.” Adults diagnosed with clinical hypothyroidism are treated with a daily fasting 48 dose of 1.6-1.8 μg/kg of levothyroxine. Every 4-6 weeks, their thyroid function is evaluated until the 49 TSH falls within the reference ranges (0.4-4.0 mIU/ml)]”.

In my opinion, in the present form the manuscript is unsuitable for publication.

Author Response

Reviewer 2:

The authors presented an interesting issue of mood disorders, which occur in hypothyroid women despite a proper treatment with levothyroxine. However, I have several doubts which make it impossible to assess whether the obtained results are true for the analyzed groups. The inclusion criteria for the Control Group included “an absence of thyroid disorders (clinical or subclinical hypothyroidism, clinical or subclinical hyperthyroidism)” and the authors claimed that all patients with abnormal TSH level within the previous 6 months had been excluded. However, no data on the TSH or thyroid hormone levels are available for this group. Were any thyroid tests performed in this group? Did all of the patients really have normal TSH within 6 months before inclusion?

First of all, thank you for your comments and suggestions to improve the manuscript. Regarding your question, serum TSH levels of all women in our study were tested in the previous 6 months. These levels were checked in order to confirm that they were within the normal reference range (0.4–4.0 mIU/L). Nevertheless, only TSH levels of the case group were collected because our aim was to assess the prevalence of mood disorders in levothyroxine-treated hypothyroid women.This is a limitation of our study that has been included in the corresponding section. Nevertheless, it will be taken into account for future studies.

Why weren`t the thyroid tests performed at the time of inclusion? How were the subclinical disorders assessed without the thyroid hormone levels? All these questions mean that the actual characteristics of the studied groups may be different from those assumed by the authors. It is impossible to obtain reliable results of the comparison without application of reliable diagnostic procedures allowing to clearly distinguish a group of healthy people and a case group.

Women were selected with a blood test of the previous 6 months, so it was considered unnecessary to repeat it at the beginning of the study. The reasons were simple: it simplified the study and reduced costs. The absence of subclinical hypothyroidism was diagnosed with the following criteria: 1) non-existence of previous diagnosis of thyroid disorder by a medical specialist; 2) normalized TSH levels in the last 6 months; 3) no hypothyroidism symptoms and 4) no family history of clinical and subclinical hypothyroidism.

According to Cooper [1], between 4 and 20% of population presents subclinical hypothyroidism, depending on the cutoff. 90% of them reports TSH levels between 4 and 10 mIU/ml [2]. This means that only 10% of the first group (which is, 0.4-2% of the total sample) could suffer from subclinical hypothyroidism with normal TSH levels and without symptomatology.

Nevertheless, there could be a woman without symptoms and TSH levels within the reference range. Evidence indicates that there may be a possible relation between subclinical hypothyroidism and the presence of anxiety and depression [3], therefore, removing a case of subclinical hypothyroidism would increase the differences between case and control group, strengthening our results.

Moreover, the authors did not provide any data on the thyroid hormone levels in the analyzed case patients. There is only an information that the TSH levels were within the normal range and that only patients with treated overt hypothyroidism were included into the study. Mood disorders are frequently present in patients and many additional factors may influence the occurrence of depression and anxiety. The authors analyzed several of these factors but the results of the whole analysis cannot be accepted without the clear group selection

Mean TSH and standard deviation [2.43 (SD 1.07)]  of case group are included in table 1. In order to make information more accessible, the data have been added in section 3.1. Furthermore, Appendix A includes a Pearson correlation between TSH levels and scores in anxiety and depression of case group. No significative results were found.

As we pointed out in the limitations section, TSH levels were tested in all the sample, but not collected from controls. We examined the analysis of the last six months and verified the absence of new symptoms. The filters to preserve the anonymity of participants prevent us from matching data and subjects in order to conduct a comparison between groups.

If the authors are able to provide the lacking data and the group re-analysis, I will be ready to review the re-submission (after extensive language editing by a native speaker).

The manuscript has been checked by a native speaker. We expect that all doubts are clarified.

The Introduction should contain essential literature-based background for the study. The obvious basic knowledge should not be included [e.g.” Adults diagnosed with clinical hypothyroidism are treated with a daily fasting 48 dose of 1.6-1.8 μg/kg of levothyroxine. Every 4-6 weeks, their thyroid function is evaluated until the 49 TSH falls within the reference ranges (0.4-4.0 mIU/ml)]”.

The introduction has been checked and basic knowledge has been removed.

References:

[1] Cooper DS, Biondi B. Subclinical thyroid disease. Lancet 2012;379:1142- 54. 10.1016/S0140-6736(11)60276-6. pmid:22273398.

[2] Rodondi N, den Elzen WP, Bauer DC, et al. Thyroid Studies Collaboration. Subclinical hypothyroidism and the risk of coronary heart disease and mortality. JAMA 2010;304:1365-74. 10.1001/ jama.2010.1361. pmid:20858880.

[3] Zhao, T.; Chen, B.B.; Zhao, X.M.; Shan, Z.Y. Subclinical hypothyroidism and depression: a meta-analysis. Transl Psychiatry, 2018, 8(1), 239.

Reviewer 3 Report

In this case-control study the authors show that hypothyroid women under levothyroxine are still at increased risk of anxiety and depression compared to euthyroid controls.

Major comments. In the whole manuscript the authors refer to the control group as “hypothyroid women” despite levothyroxine treatment. This is confusing, because the reader might think that they were still hypothyroid despite levothyroxine replacement. Please clarify. Table 4. Obese/overweight women under levothyroxine had not an increased risk of anxiety or depression, contrary to the control group. Also, in the study group, obesity had a protective, although insignificant, effect. Concerning depression, the odds ratio was very high in the control group. Please comment. I suggest to correlated anxiety and depression scores with TSH and FT4 levels.

Minor comments

The paper needs a careful proofreading as English language is poor in some points.

“The aims of this study are to assess the prevalence …” Please use the past tense. Also, this sentence is confusing (are not women without hypothyroidism, euthyroid women?). Please amend. Results. Do the authors refer to hypothyroid women treated under levothyroxine or untreated hypothyroid women? Introduction, line 34. Hypothyroidism is the most common thyroid dysfunction in the population. Introduction, line 46. The word “associated” is repeated twice. Introduction, line 47. “the numbers”. Please amend. Introduction, line 49. “1.6-1.8 μg/kg levothyroxine”. Introduction, line 54. “continue”. Introduction, line 56-57. Please rephrase. Introduction, lines 76-77. “in levothyroxine-treated hypothyroid women”. Introduction, line 77 and following lines. I suggest replacing “moods” with “mood” throughout the manuscript. Introduction, lines 77-78. Please rephrase, as this sentence is confusing. Introduction, line 85. I suggest deleting “from”. Introduction, line 89 and following lines. “The exclusion criteria were to have been diagnosed with subclinical hypothyroidism, to have high serum TSH despite replacement therapy, to be pregnant or lactating, to have serious…” Introduction, line 96 and following lines. The exclusion criteria were having a family history … , having an abnormal TSH, being on drugs that interfered with thyroid function …, being pregnant ….” Materials and Methods. How long were patient taking levothyroxine? This is important, as some clinical parameters normalize after a longer time. Please clarify Materials and Methods, line 121. Was BMI “measured” by administering a questionnaire? Please clarify. Materials and Methods, line 124. “TSH was measured beforehand as was the daily intake of levothyroxine”. Please clarify. Table 2. “Prevalencia”. Please amend. Table 2. Row 4. A parenthesis is missing. Discussion, line 239. “Recently, it has been hypothesized an influence of polymorphism of …” Discussion, line 241. Please replace “active hormone” with T3. Discussion, line 246. Please define the abbreviation “HRT”. Strengths and limitations, line 281. The wide age of the control group is a limitation, not a strenght, considering that depression risk significantly changes across age decades, as the authors state in the Discussion section (lines 256-258). Conclusion, line 288. “was not noticeable in hypothyroid women under levothyroxine”.

As the authors have mentioned brain hypothyroidism, I expected to find something on combination therapy (T4+T3) on mood (see this recent review on this topic Thyroid Res. 2018 Jan 17;11:1. doi: 10.1186/s13044-018-0045-x). The authors failed to mention that psychiatric disturbances may stem also from iatrogenic hypothyroidism (see Expert Opin Drug Saf. 2013 Nov;12(6):865-72. doi: 10.1517/14740338.2013.823397).

Author Response

Reviewer 3:

In the whole manuscript the authors refer to the control group as “hypothyroid women” despite levothyroxine treatment. This is confusing, because the reader might think that they were still hypothyroid despite levothyroxine replacement. Please clarify.

First of all, thank you for your comments and suggestions to improve the manuscript. The nomenclature has been change in order to avoid possible doubts.

Table 4. Obese/overweight women under levothyroxine had not an increased risk of anxiety or depression, contrary to the control group. Also, in the study group, obesity had a protective, although insignificant, effect. Concerning depression, the odds ratio was very high in the control group. Please comment.

Both suggestions have been included in the conclusions section.

I suggest to correlated anxiety and depression scores with TSH and FT4 levels.

The correlation with TSH levels was performed but, since it was not significative, it was removed from the article and included as an appendix -Appendix A-. Regarding FT4 levels, these data was not collected and, consequently, this correlation cannot be performed. Although we can obtain this information, the filters to preserve the anonymity of participants prevent us from matching data and subjects in order to conduct a comparison between groups. This is a limitation of our study that has been included in the corresponding section. Nevertheless, it will be taken into account for future studies.

The paper needs a careful proofreading as English language is poor in some points.

The translation has been revised.

“The aims of this study are to assess the prevalence …” Please use the past tense. Also, this sentence is confusing (are not women without hypothyroidism, euthyroid women?). Please amend.

It has been reformulated by a native speaker.

Results. Do the authors refer to hypothyroid women treated under levothyroxine or untreated hypothyroid women?

It has been corrected in all the manuscript to avoid confusion.

Introduction, line 34. Hypothyroidism is the most common thyroid dysfunction in the population.

It has been corrected.

Introduction, line 46. The word “associated” is repeated twice.

It has been corrected.

Introduction, line 47. “the numbers”. Please amend.

It has been corrected.

Introduction, line 49. “1.6-1.8 μg/kg levothyroxine”.

It has been removed.

Introduction, line 54. “continue”.

It has been corrected.

Introduction, line 56-57. Please rephrase.

It has been corrected.

Introduction, lines 76-77. “in levothyroxine-treated hypothyroid women”.

It has been corrected in all the manuscript to avoid confusion.

Introduction, line 77 and following lines. I suggest replacing “moods” with “mood” throughout the manuscript.

It has been corrected in all the manuscript.

Introduction, lines 77-78. Please rephrase, as this sentence is confusing.

It has been corrected.

Introduction, line 85. I suggest deleting “from”.

It has been corrected.

Introduction, line 89 and following lines. “The exclusion criteria were to have been diagnosed with subclinical hypothyroidism, to have high serum TSH despite replacement therapy, to be pregnant or lactating, to have serious…”

It has been corrected.

Introduction, line 96 and following lines. The exclusion criteria were having a family history … , having an abnormal TSH, being on drugs that interfered with thyroid function …, being pregnant ….”

It has been corrected.

Materials and Methods. How long were patient taking levothyroxine? This is important, as some clinical parameters normalize after a longer time. Please clarify

In our study, one of the inclusion criteria for the case group were at least 6 months of levothyroxine treatment and normalized TSH (section 2.1). The total length of the treatment was not collected.

Materials and Methods, line 121. Was BMI “measured” by administering a questionnaire? Please clarify.

During a doctor's appointment the weight and height of each patient was recorded. It has been corrected. BMI was calculated with these data.

Materials and Methods, line 124. “TSH was measured beforehand as was the daily intake of levothyroxine”. Please clarify.

It has been corrected.

Table 2. “Prevalencia”. Please amend.

It has been corrected.

Table 2. Row 4. A parenthesis is missing.

It has been corrected.

Discussion, line 239. “Recently, it has been hypothesized an influence of polymorphism of …”

It has been corrected.

Discussion, line 241. Please replace “active hormone” with T3.

It has been corrected.

Discussion, line 246. Please define the abbreviation “HRT”.

It has been corrected.

Strengths and limitations, line 281. The wide age of the control group is a limitation, not a strenght, considering that depression risk significantly changes across age decades, as the authors state in the Discussion section (lines 256-258).

Following your suggestions, the text has been removed from that section.

Conclusion, line 288. “was not noticeable in hypothyroid women under levothyroxine”.

It has been corrected.

As the authors have mentioned brain hypothyroidism, I expected to find something on combination therapy (T4+T3) on mood (see this recent review on this topic Thyroid Res. 2018 Jan 17;11:1. doi: 10.1186/s13044-018-0045-x).

Thank you for your suggestion. It has been included.

The authors failed to mention that psychiatric disturbances may stem also from iatrogenic hypothyroidism (see Expert Opin Drug Saf. 2013 Nov;12(6):865-72. doi: 10.1517/14740338.2013.823397).

We consider this as an interesting suggestion and, therefore, it has been included in the introduction.

Round 2

Reviewer 2 Report

The authors have addressed all my remarks. Several minor corrections should be introduced before publication:

Line 29-30 “Although levothyroxine is a good treatment for the symptoms of hypothyroidism, it is not enough to prevent depression and anxiety by itself”.

This sentence suggests that for all women treated with L-T4, this therapy is insufficient to prevent mood disorders. Such a conclusion is obviously false. It is known that L-T4 treatment improves the mental state and mood disorders in most patients as compared to the pre-treatment condition. Please correct the statement using e.g. “Levothyroxine is a good treatment for the symptoms of hypothyroidism, it may be not enough to prevent development or persistence of depression and anxiety by itself”.

All abbreviations should be explained in the main text where used for the first time. The Authors should not assume that all readers know the meaning of all abbreviations. I have not found explanations of many of them, including HADS (explained in line 106, first used in line 56), BDI-II, TSH etc. Line 84 is “diagnosed of primary hypothyroidism”. Should it be “diagnosis of …” or “diagnosed primary hypothyroidism”?? Line 85 – “normalized TSH levels (0.4-4-0 mIU/ml).The exclusion criteria were subclinical hypothyroidism”. A space is lacking here. Line 109 – Shall it really be “anxiety –odd items–and depression –even items–.” Did the Authors mean “anxiety – odd items and depression – even items.”? Line 167 – “Levothyroxine-treated hypothyroid women and normalized TSH (case group)” Did the Authors mean “…women with normalized TSH (case group)”? Lines 242-245 - please adjust the font size. Not all of the References are presented according to the instruction for authors. Please correct it.

Author Response

The authors have adressed all my remarks. Several minor corrections should be introduced before publication.

First of all, thank you for your comments and suggestions to improve the manuscript.

Line 29-30 “Although levothyroxine is a good treatment for the symptoms of hypothyroidism, it is not enough to prevent depression and anxiety by itself”. This sentence suggests that for all the women treated with L-T4, this therapy is insufficient to prevent mood disorders. Such a conclusion is obviously false. It is known that L-T4 treatment improves the mental state and mood disorders in most patients as compared to the pre-treatment condition. Please correct the statement using e.g. Levothyroxine is a good treatment for the symptoms of hypothyroidism, it may be not enough to prevent development or persistence of depression and anxiety by itself”.

Following your suggestion, the text has been corrected.

All abbreviations should be explained in the main text where used for the first time. The authors should not assume that all readers know the meaning of all abbreviations. I have not found explanations of many of them, including HADS (explained in line 106, first used in line 56), BDI-II, TSH, etc.

It has been corrected.

Line 84 is diagnosed of primary hypothyroidism. Should it be diagnosis of” or “diagnosed primary hypothyroidism”??

It has been corrected.

Line 85 - “normalized TSH levels (0.4-4.0 mIU/ml).The exclusion criteria were subclinical hypothyroidism. A space is lacking here.

It has been corrected.

Line 109 - Shall it really be anxiety - odd items - and depression - even items -. Did the authors mean anxiety - odd items and depression - even items”?

It has been corrected.

Line 167 - “Levothyroxine-treated hypothyroid women and normalized TSH (case group)Did the authors mean “…women with normalized TSH (case group)”?.

It has been corrected.

Lines 242-245 - please adjust the font size.

It has been corrected.

Not all the References are presented according to the instructions for authors. Please correct it.

It has been corrected.

Reviewer 3 Report

Lines 83-86. The inclusion criteria for the case group (women with primary hypothyroidism) were: overt hypothyroidism treated with levothyroxine for at least 6 months, and normalized TSH levels (0.4-4-0 mIU/ml). The exclusion criteria were subclinical hypothyroidism, high serum TSH …” Lines 91-95. In order not to repeat what is written some lines above, I suggest stating that exclusion criteria were the same of the study group. Table 1. The footnote * is redundant. Lines 239-240. “The type 2 deiodinase converts T4 into T3 in the brain and the hypophysis, for instance [22].” Lines 301. Please delete “of”. Line 304. “carried out on a anxiety and depression”. Line 305. Please delete “case group”. In the manuscript I suggest replacing “women without hypothyroidism” or “non-hypothyroid women” with “euthyroid women”.

Author Response

Lines 83-86. The inclusion criteria for the case group (women with primary hypothyroidism) were: overt hypothyroidism treated with levothyroxine for at least 6 months, and normalized TSH levels (0.4-4.0 mIU/L). The exclusion criteria were subclinical hypothyroidism, high serum TSH…”

First of all, thank you for your comments and suggestions to improve the manuscript. Primary has been preferred to overt in order to avoid possible confusion since overt could suggest that symptoms are not under treatment. Everything else has been rewritten.

Lines 91-95. In order not to repeat what is written some lines above, I suggest stating that exclusion criteria were the same of the study group.

It has been corrected.

Table 1. The footnote * is redundant.

It has been removed.

Lines 239-240. “The type 2 deiodinase converts T4 into T3 in the brain and the hypophysis, for instance [22]”.

It has been corrected.

Lines 301. Please delete of”.

It has been corrected.

Line 304,  carried out on a anxiety and depression”.

It has been corrected.

Line 305. Please delete case group”.

It has been corrected.

In the manuscript I suggest replacing women without hypothyroidism” on “non-hypothyroid womenwith euthyroid women”.

Since both groups–case and control–are composed by women with normalized TSH levels (TSH levels 0.4-4.0 mIU/L) we prefer the original terminology. This will avoid confusion for readers.